# Diet, Digestion, and the Dietitian: A Survey of Clinicians’ Knowledge, Attitudes and Practices to Advance the Treatment of Gastrointestinal Disturbances in Individuals with Anorexia Nervosa

**DOI:** 10.3390/jcm11195833

**Published:** 2022-09-30

**Authors:** Madeline L. West, Caitlin McMaster, Claire L. Young, Mohammadreza Mohebbi, Susan Hart, Heidi M. Staudacher, Amy Loughman, Anu Ruusunen, Tetyana Rocks

**Affiliations:** 1Food & Mood Centre, IMPACT—The Institute for Mental and Physical Health and Clinical Translation, School of Medicine, Barwon Health, Deakin University, Geelong 3220, Australia; 2Children’s Hospital Westmead Clinical School, Faculty of Medicine and Health, The University of Sydney, Westmead 2145, Australia; 3Eating and Nutrition Research Group, School of Medicine, Western Sydney University, Campbelltown 2560, Australia; 4Faculty of Health, Biostatistics Unit, Deakin University, Geelong 3216, Australia; 5Nutrition Services, St Vincent’s Health Network, Darlinghurst 2010, Australia; 6Institute of Public Health and Clinical Nutrition, University of Eastern Finland, 70211 Kuopio, Finland; 7Department of Psychiatry, Kuopio University Hospital, 70210 Kuopio, Finland

**Keywords:** anorexia nervosa, dietetic treatment, nutritional rehabilitation, gastrointestinal disorder, gastrointestinal disturbance

## Abstract

Despite advances in treatment of anorexia nervosa (AN), current therapeutic approaches do not fully consider gastrointestinal disturbances (GID), often present in AN. Addressing GID, both symptoms and disorders, is likely to improve treatment adherence and outcomes in people with AN. GID are complex and are linked to a range of factors related to eating disorder symptomology and can be impacted by nutritional treatment. It is not known which dietetic practices are currently used to address GID in AN. Therefore, this survey aimed to explore the perceived knowledge, attitudes, and practices (KAP) of Australian dietitians treating AN and co-occurring GID. Seventy dietitians participated by completing an online survey. Knowledge scores were calculated based on correct responses to knowledge items (total: 12 points); and two groups were generated: higher knowledge (≥10 points, *n* = 31) and lower knowledge (≤9 points, *n* = 39). A greater proportion of dietitians with higher knowledge recognized the role of GID in pathogenesis of AN (*p* = 0.002) and its impact on quality of life (*p* = 0.013) and screened for GID (*p* ≤ 0.001), compared with those with lower knowledge. These results suggest that attitudes and practices toward patients presenting with AN and GID differ depending on level of knowledge. This may have important implications for treatment outcomes for individuals with AN and GID.

## 1. Introduction

Gastrointestinal disturbances (GID) are common experiences for individuals living with anorexia nervosa (AN) [1,2]. AN has a low recovery rate and the highest mortality rate of all psychiatric disorders [3]. Thus, advances in treatment to improve outcomes, including treatment of GID, are vital. Almost always GID have no underlying structural cause and can present as functional symptoms [4], such as irritable bowel syndrome (IBS), abdominal pain or constipation. The relationship between GID and AN is complex, and the aetiology is unclear. The presence of gastrointestinal (GI) problems during childhood is a risk factor for the later emergence of AN [5], so some individuals may experience GID before the onset of AN. Whereas for others, the experience of GID may present as a culmination of disordered eating behaviours and psychological distress [6,7]. However, it is well documented that whilst GID presentation may change or improve during inpatient care [8,9], for many people symptoms persist after treatment for AN [10]. Therefore, greater consideration is needed regarding how GID may be targeted in the treatment of AN to improve outcomes.

GI-related problems are the leading cause of hospital admission in young people with eating disorders (ED) [11] and may be related to worse quality of life [12]. Additionally, GID also contribute to several aspects of AN, including poor body image where an individual’s attention is disproportionately focussed on the abdomen [13], and feelings of fullness and bloating can trigger body image distress [14]; poor adherence to treatment as GID have been identified as a reason for treatment refusal [15]; and, it has been hypothesized that GID may also contribute to overall treatment success [6,16]. Commonly, GID are believed to resolve with weight restoration, and specific treatment approaches have not been rigorously tested. Although their efficacy is unclear, medications including prokinetic agents, smooth muscle relaxants and laxatives may be used to manage GID in AN, providing symptom relief to facilitate adherence to nutritional treatments [17]. Emerging evidence suggests the use of brain–gut psychotherapies [18], and the need for novel renourishment techniques that address the underlying altered GI physiology [19]. To inform appropriate treatments, the effectiveness of current GID treatment strategies in people with AN must be evaluated.

In general, treatment for AN primarily involves medical stabilisation, nutritional rehabilitation, and weight restoration, accompanied by psychological support [20]. GID presence and severity can be influenced by a range of biopsychosocial factors, including psychological distress, mood, malnutrition, dietary intake, and exercise [4]. Hence, the various components of multidisciplinary treatment for an ED may also influence GID, but the relationships between these components and the extent to which they may influence GID have not yet been investigated. Of particular interest is the effect of dietary treatment on GID in people with AN.

Addressing GID is within dietetic scope of practice [21]; however, it is unclear which dietetic treatments are currently used to address GID in people with AN. Individuals with AN are diagnosed with IBS more often than the general population [22]. In otherwise healthy individuals with IBS, dietary modifications such as the low FODMAP (fermentable oligo-, di-, monosaccharides and polyols) diet, which restricts specific foods high in fermentable carbohydrates for a short period of time, has good efficacy for reducing IBS symptoms [23]. However, such exclusion diets are often not appropriate for individuals with AN and co-occurring IBS [24], who are working toward overcoming rigid food rules and food restrictions [20], but whether they are used in practice is unknown. Functional constipation is also commonly experienced by individuals with AN [25,26]. For constipation, increased fibre intake and adequate fluid are often recommended [27]. Increasing dietary fibre in individuals with AN needs to be implemented with caution as high fibre foods are usually not energy dense and are likely to contribute to the sensation of fullness, both of which could compromise meeting nutrient targets for AN recovery [28,29]. Dietetic management of GID in individuals with AN has not been investigated and it is unclear what techniques, if any, are being utilised in practice.

Given the scant literature exploring dietetic management of GID in individuals with AN, little is known of dietitians’ knowledge and attitudes around the topic, or what dietetic therapies are being used. Therefore, this study aimed to assess the knowledge, attitudes, and practices of dietitians treating individuals with AN and co-occurring GID, and to explore the relationship between knowledge and attitudes, and knowledge and practices.

## 2. Materials and Methods

### 2.1. Study Design

This cross-sectional online survey used convenience sampling to recruit Australian Dietitians in February and March 2022.

### 2.2. Participants

Participants were eligible for registration as an Accredited Practising Dietitian (APD), and had current or recent experience treating patients with a diagnosis of AN. The study was advertised to dietitians via the Dietitians Australia weekly newsletter and the Australian and New Zealand Academy of Eating Disorders website. Advertisements were also posted on the study team’s personal and affiliate social media pages.

Dietitians were provided with a plain language statement explaining the study procedures. Participants were screened online and gave their consent digitally if they were eligible and wished to participate in the study. Participation in the study was voluntary and anonymous. Deakin University’s Human Ethics Advisory Group, Faculty of Health, approved the study protocol (HEAG-H 119_2021) and the study was prospectively registered with Open Science Framework (OSF) registry (registration DOI: https://doi.org/10.17605/OSF.IO/YNTW9). A target sample size of *n* = 70 dietitians was determined based on a conservative estimate that 5% of the Dietitians Australia eating disorders interest group (*n* = 1400 members of a total *n* = 7990 dietitians) would agree to participate.

### 2.3. Survey Instrument

We developed the 32-item Dietitians Anorexia Nervosa Gastrointestinal Knowledge, Attitudes, and Practices Survey (ANGI-KAPS). This was guided by published knowledge, attitude and practices (KAP) literature [30,31] as well as previous studies of dietitian KAPs [32,33], and other health professionals KAPs of GID [34] or EDs [35]. The ANGI-KAPS tool was designed to assess the knowledge, attitudes and practices of ED dietitians treating individuals with AN and GID. The survey was developed in a two-step process. In brief, step one was instrument design, performed by: (i) a review of the literature and validated tools to identify topic areas of interest, (ii) item generation, and (iii) piloting by dietitians with expertise in ED (*n* = 3) and GID (*n* = 2). The pilot panel were asked to rate each item as satisfactory or unsatisfactory, and to provide comments for items rated unsatisfactory [30]. Items that received any number of unsatisfactory ratings were re-considered by the research team, as were comments and suggestions provided. Following the pilot panel review, the research team revised the instrument to 33 items (9 knowledge, 11 attitudes, 8 practices, 3 KAP gut microbiome items). Step two was evaluation of face validity and content validity, conducted in collaboration with a panel of expert ED dietitians who were (1) eligible for registration as an accredited practising dietitian, (2) currently working with individuals with an ED, and (3) living in Australia. Mean years practising as a dietitian was 21.8 years (SD ± 13.02) and the mean years treating individuals with AN was 16.4 years (SD ± 9.13). Participants reviewed each item in the ANGI-KAPS tool and were asked to rate the items for relevance, essentiality, and clarity [36]. At the completion of the validation step, the final instrument contained 32 items (9 knowledge items, 11 attitude items, 9 practice idents, and 3 KAP gut microbiome items). The final version of the survey is included as Appendix A.

The ANGI-KAPS survey first collected demographic and clinical experience data from the dietitians. This included workplace setting, years of practice and additional training in psychology or GID. Proportion of work time spent with individuals with AN, and years treating AN were also collected. Knowledge items assessed knowledge about the intersection of AN and GID. Knowledge items comprised a combination of multiple choice, true-false and Likert scale questions and correct answers were calculated for a knowledge score (1 point for each correct answer). Attitude and practice items were multiple choice or Likert scale (agreeability: 1 = strongly disagree, 2 = somewhat disagree, 3 = neither agree nor disagree, 4 = somewhat agree, 5 = strongly agree; or frequency: 1 = none of the time, 2 = some of the time, 3 = not sure, 4 = most of the time, 5 = all of the time) questions.

Data were collected anonymously via the secure Deakin University Qualtrics^TM^ platform.

### 2.4. Statistical Analysis

Descriptive statistics were used for participant demographics and knowledge, attitudes and practices. A knowledge score was calculated by summing the number of correct answers to knowledge items (9 items, maximum score of 12). Participants were grouped based on their level of knowledge (higher knowledge or lower knowledge), determined by a median split. Individuals with scores ≥ 10 points were allocated to the higher knowledge group, and those with scores ≤ 9 were allocated to the lower knowledge group.

Chi-square tests assessed the difference in responses to attitude and practice items between groups with higher and lower knowledge. Odds ratios and 95% confidence intervals were calculated to aid the interpretation of chi-square tests. The difference in Likert scale responses to attitude and practice items was assessed by estimating the mean difference and 95% confidence intervals between knowledge groups via independent samples t-tests. *p*-values of < 0.05 were considered statistically significant, and Cohen’s D of d = 0.2 was considered a ‘small’ effect size, 0.5 was considered a ‘medium’ effect size and 0.8 was considered a ‘large’ effect size [37]. Statistical analyses were completed using STATA [38].

## 3. Results

Seventy dietitians (91% female) completed the survey. Participants were grouped based on higher (≥10 points, *n* = 31) and lower knowledge (≤9 points, *n* = 39) of AN and GID, as measured by the survey. The median knowledge score was 9 (1–12). There were no differences between groups for years of practising as a dietitian or years of experience treating individuals with AN. More dietitians worked in private practice in the higher knowledge group compared with the lower knowledge group (77% vs. 44%, *p* = 0.004). A significantly greater proportion of dietitians in the higher knowledge group had received formal training in the treatment of AN (*p* = 0.031) and formal training in the treatment of co-occurring AN and GID (*p* = 0.011) compared with the lower knowledge group, see Table 1.

### 3.1. Attitudes

Responses to attitudes items are presented in Table 2. Compared to the lower knowledge group, a greater proportion of dietitians in the higher knowledge group identified GID as playing a role in the pathogenesis of AN (77% vs. 41%, *p* = 0.002, OR: 4.93 (1.71, 14.17)) and impacting on quality of life (100% vs. 82%, *p* = 0.013). Further, a greater proportion of dietitians in the higher knowledge group believed that GID can encourage ED behaviours (97% vs. 77%, *p* = 0.018, OR: 9.00 (1.07, 75.51)), complicate treatment (97% vs. 79%, *p* = 0.032, OR: 7.74 (0.091, 65.71)) and be a conditioned response to feared foods (68% vs. 46%, *p* = 0.015, OR: 3.36 (1.25, 9.06)). Those in the higher knowledge group were also more likely to strongly agree that patients perceive GID as a barrier to achieving recovery compared with the lower knowledge group (4.52 (0.63) vs. 4.03 (0.84) *p* = 0.009, *d* = −0.65 (−1.13, 0.16)).

### 3.2. Practices

Several differences were observed between higher and lower knowledge groups in relation to screening practices and treatments implemented (Table 3). Dietitians in the higher knowledge group were more likely to frequently screen patients for GIDs than the lower knowledge group, with a significant and large effect size (4.19 (1.08); 3.15 (1.27) *p* ≤ 0.001, *d* = −0.88 (−1.37, −0.38)). Treatment practices more likely to be adopted by dietitians in the higher knowledge group compared with the lower knowledge group included suggesting the use of anti-diarrheal agents (22% vs. 3%, *p* = 0.009; OR: 11.08 (1.28, 95.79)), recommending use of gut-directed hypnotherapy (52% vs. 18%, *p* = 0.003; OR: 4.88 (1.66, 14.35)), provision of reassurance (90% vs. 66%, *p* = 0.027; OR: 5.38 (1.08, 26.92)), and suggesting consultation with a psychologist (55% vs. 23%, *p* = 0.009; OR: 3.92 (1.38, 11.15). In general, dietitians in the higher knowledge group felt more confident approaching a patient with co-occurring GID and AN compared with lower knowledge, with a medium effect size (3.62 (0.091) *p* ≤ 0.006, *d* = −0.69 (−1.17, −0.20)). However, there was no difference in confidence in the ability to treat co-occurring GID and AN between groups (3.84 (0.90); 3.51 (0.85) *p* = 0.13, *d* = −0.37 (−0.85, 0.10)).

## 4. Discussion

This is the first study to investigate dietitians’ knowledge, attitudes and practices about GID in individuals with AN. Overall, dietitians with higher knowledge, as measured by our survey, had greater insight into the complexity of GID in AN, and more confidence approaching GID in individuals with AN. Together, these attributes of dietitians with higher knowledge may encourage the provision of holistic and supportive care.

Firstly, attitudes that differed between dietitians with higher and lower knowledge of GID and AN, related to the impact of GID on an individual’s disorder experience and its impact on quality of life. These attitudes may be important to establish a shared understanding of the individuals’ experiences and lay the foundations for a strong therapeutic relationship [39]. Healthcare providers often report challenges in developing therapeutic relationships with individuals with EDs [40,41]. Frequent relapses [42], treatment ambivalence [43], disorder complexity and feeling helpless [44] are often reported as barriers. The presence of GID may add further complexity as stigma and shame are frequent experiences for individuals with functional GI disorders [45] and also for treatment seeking in AN. In such individuals, an effective therapeutic relationship improves patient satisfaction, symptom management and clinical outcomes [46]. Additionally, therapeutic relationships that are built on trust and honest communication may reduce symptom-related anxieties [47]. This may translate to individuals with EDs as heightened visceral sensitivity and GI-specific anxiety are often present [29,48]. Given that dietitians with higher knowledge scores on our survey were more understanding of the interference of GID on AN and quality of life, this may help to strengthen the clinician-patient relationship, which, in turn, can have a positive influence on health outcomes [49].

Secondly, dietitians who scored higher on our knowledge score were more likely to engage in practices that seek to identify GID, communicate information about GID and recommend specific treatments to their patients. Considerations relevant to GID are outlined in practice and training standards for ED dietitians in Australia [50] which include obtaining ‘history of gastrointestinal signs and symptoms in the context of the eating disorder and comorbidities’. Results from the current survey suggest that dietitians who scored lower on our knowledge score may not be aware of the need to investigate for GID; less than 40% of total participants reported screening for GID using tools listed in our survey. Several participants reported using their own clinical judgement, or personally developed tools to identify GID, which may suggest a lack of knowledge of validated tools for this purpose. Participants reported using a range of practices to address GID, including referral to gastroenterologists, advice on meal timing and pre- or post-meal activities. Differences between higher and lower knowledge groups emerged in the suggestions of anti-diarrhoeal agents and gut-directed hypnotherapy. Dietitians who scored higher on our knowledge score were more likely to suggest the use of gut-directed hypnotherapy, which works by targeting the stress response to GID [18]. This may suggest heightened awareness of the role of the gut-brain axis in GID, and AN more broadly. Additionally, significantly fewer dietitians in the lower knowledge group, compared with the higher knowledge group, believed that GID education should be provided by the dietitian. Nutritional counselling for managing symptoms such as constipation, bloating and reflux has been identified as an essential component of outpatient dietetic treatment by ED dietitians, clinicians, consumers and carers [51]. However, our results suggest a significant portion of dietitians may not believe this is part of a dietitian’s role. Furthermore, the assessment and counselling for managing GID is not featured in ED treatment guidelines [52]. While the ANZAED Dietetic Practice Standards suggest GID as a topic that may be relevant for education [50], the Royal Australian and New Zealand College of Psychiatrists note that reassurance around GID is a required action [20]. This gap in guidelines may explain differences in practice and highlights the need for a specific framework to guide assessment and intervention.

Findings from this study suggest practice and attitudes of dietitians might differ depending on level of knowledge of the role GID might play in ED. Knowledge has been identified as an important component of dietetic clinical decision-making [53]. Additionally, knowledge is important for clinician adherence to treatment guidelines [54]. Importantly, many dietitians report the need for further training in the treatment of ED [55,56], and findings from the present survey suggest around half of the dietitians surveyed could improve their knowledge of GID in AN. The provision of training and education for identifying and managing GID in AN may improve the capacity of dietitians to deliver tailored therapy to individuals with AN, from screening to treatment. The inclusion of GID within dietetic and other clinical ED guidelines would also formalize expected knowledge and support the widespread inclusion of GID management strategies for AN into dietetic and professional training.

Implementation of routine assessment of GID within the dietetic treatment of AN offers the potential to improve treatment outcomes. Although the efficacy of current dietetic treatment for GID in AN have not been evaluated, malnutrition has a direct impact on the physiology of the GI tract, including altering motility [57]. Furthermore, engagement in ED behaviours, such as restriction and purging are directly related to GID presence [1]. Therefore, normalizing eating behaviours and hence, food consumption, has consequences for both the function of the GI tract and GID, and should be considered when planning nutritional treatment. To improve response to treatment, dietetic recommendations should be tailored to ensure practices are not exacerbating GID.

The strengths of this study include the reporting of novel data investigating dietitians’ knowledge, attitudes and practices in treating individuals with AN and GID. The data from the present study provide vital insights into attitudes and practices of dietitians managing individuals with AN, specifically relating to GID. Furthermore, the survey tool was developed and validated especially for the purposes of this study, and so it is relevant to the target population of Australian dietitians. The sample of dietitians recruited for this study is representative of the largely female workforce in Australia where approximately 95% of dietitians identify as female [58]. However, a limitation of this study is that the survey did not collect other demographic characteristics to enable a more thorough comparison to the Australian workforce, such as age, weekly hours worked, Aboriginal and Torres Strait Islander Status or country of birth.

Further limitations of this study include the small sample size. Whilst the target sample size was achieved, surveying a greater number of dietitians would have improved the statistical power. Additionally, knowledge group cut-off scores were determined a priori (higher knowledge group scores ≥ 7; lower knowledge scores ≤ 6), however, after reviewing the distribution of data, which was skewed toward higher knowledge scores, the study team decided it more appropriate to use a median split. Lastly, limitations in the survey itself must also be acknowledged. The survey is newly developed and has not been widely used. The survey questions asked participants about their usual practice but did not consider usual practices for varying levels of illness severity. Furthermore, many participants worked across multiple settings, so it is unlikely participants were drawing from the same experiences when completing the survey. Future studies may aim to capture this nuance by asking participants about how they treat GID in different settings or including case vignettes.

## 5. Conclusions

Results from this survey document Australian dietitians’ current knowledge, attitudes, and practices regarding GID in individuals living with AN. Data show that dietitians with higher knowledge, as measured by our survey, have a greater understanding of the impact of GID in AN and may provide more holistic and supportive care. Given the frequent and debilitating GID in individuals with AN, dietitians need to be knowledgeable and confident in addressing GID in this patient group, in order to optimise treatment outcomes.

## Figures and Tables

**Table 1 jcm-11-05833-t001:** Participant Characteristics.

	Higher Knowledge (*n* = 31)	Lower Knowledge (*n* = 39)	*p*-Value
Demographic	*n* (%) Mean (SD)	*n* (%) Mean (SD)	
Gender (Female)	28 (90)	36 (92)	
Years practicing as a dietitian	11.1 (8.6)	14.7 (9.7)	0.11
Years treating AN	5.9 (5.2)	8.7 (8.2)	0.11
**Settings treating AN**			
Private practice	24 (77)	17 (44)	0.004 *
General medical hospital	8 (26)	16 (41)	0.18
Specialized inpatient ED unit	6 (19)	9 (23)	0.71
Specialized inpatient psychiatric facility	1 (3)	2 (5)	0.70
Outpatient facility or program	6 (19)	12 (31)	0.28
Online/telehealth	9 (29)	6 (15)	0.17
**Workplace setting**			0.84
Urban	26 (84)	32 (82)	
Rural	5 (16)	7 (18)	
**Proportion of working time spent with individuals with AN**			0.16
0–10%	6 (19)	13 (33)	
10–25%	5 (16)	10 (26)	
25–50%	10 (32)	5 (13)	
>50%	10 (32)	11 (28)	
**Age group treating**			
Children (<16 years)	12 (39)	15 (38)	0.98
Young adults (16–18 years)	23 (74)	25 (64)	0.37
Adults (>18 years)	30 (97)	33 (85)	0.09
**Training**			
Received formal training in treating AN	28 (90)	27 (69)	0.03 *
Received formal training in psychological therapies	19 (61)	17 (44)	0.09
Received formal training in treating co-occurring AN and GI	17 (55)	9 (23)	0.01 *

* Significant differences, chi-squared test. Abbreviations: AN; anorexia nervosa, ED; eating disorder, GID; gastrointestinal disturbance.

**Table 2 jcm-11-05833-t002:** Attitudes of dietitians with higher (*n* = 31) and lower (*n* = 39) knowledge of gastrointestinal disturbances in anorexia nervosa.

Attitude Item	Higher Knowledge *n* (%)	Lower Knowledge *n* (%)	*p*-Value	^^^ OR 95% CI or ^+^ Cohen’s D (95% CI)
**In functional gastrointestinal disorders, what does ‘functional’ mean to you?**				
Symptoms likely to have a psychosomatic basis, probably representing somatization of psychological disturbance	11 (35)	11 (28)	0.52	1.40 (0.51, 3.86) ^^#^
A real GI disorder which is currently unexplained and poorly understood	20 (65)	27 (69)	0.68	0.81 (0.30, 2.20) ^^#^
**I believe GID play a role in the following aspects of AN:**				
Pathogenesis	24 (77)	16 (41)	0.002 *	4.93 (1.71, 14.17) ^^#^
Engagement in ED behaviours	30 (97)	32 (82)	0.054	6.56 (0.76, 56.54) ^^#^
Engagement in treatment	30 (97)	33 (85)	0.092	5.45 (0.62, 47.96) ^^#^
Response to treatment	25 (80)	26 (67)	0.19	2.08 (0.69, 6.34) ^^#^
Medical complications	21 (68)	22 (56)	0.33	1.62 (0.61, 4.34) ^^#^
Intestinal microbiota composition	26 (84)	27 (69)	0.16	2.31 (0.71, 7.48) ^^#^
Achieving recovery	26 (84)	25 (64)	0.065	2.91 (0.91, 9.28) ^^#^
Quality of life	31 (100)	32 (82)	0.013 *	N/A
**Within the medical team, I believe treatment of GID is the responsibility of:**				
Psychiatrist	17 (55)	14 (36)	0.11	2.17 (0.83, 5.68) ^^#^
Nurse	14 (45)	12 (31)	0.22	1.85 (0.69, 4.94) ^^#^
Physician	27 (87)	35 (90)	0.73	0.77 (0.18, 3.37) ^^#^
Dietitian	30 (97)	32 (82)	0.054	6.56 (0.76, 56.55) ^^#^
Physiotherapist	12 (39)	8 (20)	0.094	2.45 (0.85, 7.07) ^^#^
Gastroenterologist	29 (93)	33 (85)	0.243	2.64 (0.49, 14.09) ^^#^
**I believe gastrointestinal disturbances:**				
Are psychosomatic	17 (55)	14 (36)	0.11	2.17 (0.83, 5.68) ^^#^
Are a symptom of disordered eating	26 (84)	29 (74)	0.34	1.79 (0.54, 5.94) ^^#^
Are a symptom of disordered attitudes toward food and eating	17 (55)	17 (43)	0.35	1.57 (0.61, 4.06) ^^#^
Can encourage ED behaviours	30 (97)	30 (77)	0.018 *	9.00 (1.07, 75.51) ^^#^
Can complicate treatment	30 (97)	31 (79)	0.032 *	7.74 (0.91, 65.71) ^^#^
A conditioned response to feared foods	21 (68)	15 (46)	0.015 *	3.36 (1.25, 9.06) ^^#^
	Mean (SD)	Mean (SD)		
**I believe the dietitian should assist in management of GID**	4.87 (0.34)	4.64 (0.58)	0.056	−0.47 (−0.94, 0.012) ^+^
**Assessing, diagnosing and treating gastrointestinal disturbances is within my scope of practice as a dietitian**	3.77 (0.76)	3.77 (0.90)	0.98	−0.006(−0.47, 0.47) ^+^
**I believe patients perceive GID as a barrier to achieving recovery**	4.52 (0.63)	4.03 (0.84)	0.0087	−0.65 (−1.13, −0.16) ^+^
**I believe GID are a symptom of AN and will resolve over time, without specific treatment**	2.68 (1.01)	2.85 (0.96)	0.48	0.17 (−0.30, 0.64) ^+^
**I believe GID have major consequences for a patient’s quality of life**	4.81 (0.48)	4.54 (0.64)	0.057	−0.47 (−0.94, 0.014) ^+^
**If patients report GID, I modify treatment**	3.03 (1.11)	3.08 (1.16)	0.87	0.039 (−0.43, 0.51) ^+^
**I believe my view and my patients’ view of their experience with GID is generally similar?**	3.39 (1.12)	3.15 (0.99)	0.36	−0.22 (−0.70,0.25) ^+^

* Significant differences; ^^^ OR 95% CI; ^+^ Cohen’s D (95% CI). ^#^ Reference group for odds ratio is higher knowledge group. Abbreviations: AN; anorexia nervosa, ED; eating disorder, GID; gastrointestinal disturbance, N/A; not applicable. Bolded statements represent questions asked in the survey.

**Table 3 jcm-11-05833-t003:** Practices of dietitians with higher (*n* = 31) and lower (*n* = 39) knowledge of gastrointestinal disturbances in anorexia nervosa.

Practice Item	Higher Knowledge *n* (%)	Lower Knowledge *n* (%)	*p*-Value	^^^ OR 95% CI or ^+^ Cohen’s D (95% CI)
**What tool do you use to screen patients for functional GI disorders?**				
ROME	11 (35)	9 (23)	0.127	1.1 (0.41, 2.97) ^^#^
Manning	1 (3)	3 (7)	0.122	0.15 (0.018, 1.31) ^^ #^
Kruis	1 (3)	0 (0)	0.104	0.29 (0.031, 2.75) ^^ #^
**Strategies I use to address GI disturbances include:**				
Refer to gastroenterologist	21 (68)	27 (69)	0.89	0.933 (0.34, 2.57) ^^#^
Medication advice	11 (35)	9 (23)	0.25	1.83 (0.64, 5.22) ^^#^
Suggest peppermint oil	6 (19)	8 (21)	0.90	0.93 (0.29, 3.03) ^^#^
Low FODMAP diet	8 (26)	10 (26)	0.99	1.00 (0.34, 2.97) ^^#^
Exclusion of food groups	4 (13)	5 (13)	0.99	1.00 (0.25, 4.12) ^^#^
Suggest fibre supplement	17 (55)	15 (38)	0.17	1.94 (0.75, 5.06) ^^#^
Suggest probiotics	12 (39)	16 (41)	0.84	0.91 (0.335, 2.38) ^^#^
Over the counter nutrition supplements	3 (9)	4 (10)	0.94	0.94 (0.19, 4.54) ^^#^
Suggest anti-diarrhoeal agents	7 (22)	1 (3)	0.009 *	11.08 (1.28, 95.79) ^^#^
Advice on meal timing	24 (77)	31 (79)	0.83	0.88 (0.28, 2.78) ^^#^
Pre- or post-meal activities	22 (71)	21 (54)	0.14	2.10 (0.77, 5.69) ^^#^
Gut-focused hypnotherapy	16 (52)	7 (18)	0.003 *	4.88 (1.66, 14.35) ^^#^
Refer to psychologist or counsellor	19 (61)	17 (44)	0.14	2.05 (0.78, 5.35) ^^#^
Breathing techniques	15 (48)	13 (33)	0.20	1.88 (0.71, 4.94) ^^#^
Mindful eating techniques	20 (65)	18 (46)	0.13	2.12 (0.81, 5.59) ^^#^
Provide information about GID and ED	28 (90)	28 (72)	0.054	3.66 (0.92, 14.57) ^^#^
I don’t use any specific strategies	0	2 (5)	0.20	
**What type of gastrointestinal education do you provide to your patients?**				
Psychoeducation	23 (74)	20 (51)	0.073	2.63 (0.90, 7.67) ^^#^
Reassurance	28 (90)	26 (66)	0.027 *	5.38 (1.08, 26.92) ^^#^
Suggest consultation with pediatrician	6 (19)	7 (18)	0.96	1.04 (0.31, 3.50) ^^#^
Suggest consultation with psychologist	17 (55)	9 (23)	0.009 *	3.92 (1.38, 11.15) ^^#^
Suggest consultation with GP	24 (77)	24 (62)	0.23	1.45 (0.70, 3.01) ^^#^
Suggest consultation with gastroenterologist	22 (71)	22 (56)	0.29	1.75 (0.61, 5.00) ^^#^
	Mean (SD)	Mean (SD)		
**I am confident in my ability to treat co-occurring GID and anorexia nervosa**	3.84 (0.90)	3.51 (0.85)	0.13	−0.37 (−0.85, 0.10) ^+^
**I expect GID to improve with weight restoration**	2.71 (1.37)	2.13 (0.98)	0.04 *	−0.50 (−0.98, −0.017) ^+^
**Education about the relationship between GID and ED should be provided by the dietitian**	4.84 (0.45)	4.28 (0.83)	0.0013 *	−0.810 (−1.30, −0.32) ^+^
**I routinely screen patients for functional GI disorders or disturbances**	4.19 (1.08)	3.15 (1.27)	0.0005 *	−0.88 (−1.37, −0.38) ^+^
**I provide education to patients about gastrointestinal function and gut health:**	3.94 (1.06)	3.21 (1.22)	0.010 *	−0.63 (−1.12, −0.15) ^+^
**In general, I feel confident approaching a patient with co-occurring anorexia nervosa and gastrointestinal disturbances**	4.19 (0.75)	3.62 (0.91)	0.0057 *	−0.69 (−1.17, −0.20) ^^^

* Significant differences; ^^^ OR 95% CI; ^+^ Cohen’s D (95% CI). ^#^ Reference group for odds ratio is higher knowledge group. Abbreviations: AN; anorexia nervosa, ED; eating disorder, GID; gastrointestinal disturbance. Bolded statements represent questions asked in the survey.

## Data Availability

The data presented in this study are available on request from the corresponding author. The data are not publicly available due to ethical reasons.

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
