# Peer review of "Diet, Digestion, and the Dietitian: A Survey of Clinicians’ Knowledge, Attitudes and Practices to Advance the Treatment of Gastrointestinal Disturbances in Individuals with Anorexia Nervosa"

_jcm, 2022, doi:10.3390/jcm11195833_

Round 1

Reviewer 1 Report

The study evaluates the level of knowledge of 70 dieticians in Australia with regards to the importance of GID symptoms in patients with anorexia nervosa. My concerns are:

1- Authors may need to include the term "knowledge" in your title since the study is about the importance of knowledge in the pratice of the dietician

2- In the introduction, authors are trying first to expose why GID symptoms are have to be managed in patients with anorexia nervosa. They provide several studies without really making their point clear. For example, it is not clear how GID symptoms are associated with poor body image. It is not clear how much patients with GID have poor treatment adherence and hospital admission as compared with patients without GID.

3- I think that one of the goals for the introduction is to provide the reader with information about GID symptoms especially regarding what are their causes and how to manage them. Authors have named 3 causes for GID symptoms (IBS, abdominal pain and constipation) and provided a brief dietetic management for IBS and constipation. However, the idea is still mending. It is important to be more authoritative and clearer about the main causes of GID symptoms and what should be ususally suggested for patients with anorexia nervosa and GID symptoms.

4- Authors may have access to data related to Australian dieticians' age, gender, years of experience, location of practice, etc. Accordingly, a comparison between the group of responders with the group of Australian dieticians in relation to some sociodemographic parameters may give the study more power.

5- The ANGI-KAPS was not rightly described. First, the knowledge part needs to be explicated. The reader may need to know what are the questions asked in the instrument and to develop a personal judgment on it. Second, the tool's validity needs to be better discussed.   

Author Response

1- Authors may need to include the term "knowledge" in your title since the study is about the importance of knowledge in the practice of the dietician

Thank you for this suggestion, we have updated the title to more accurately describe the study: Diet, digestion, and the dietitian: a survey of clinicians’ knowledge, attitudes and practices to advance the treatment of gastrointestinal disturbances in individuals with anorexia nervosa.

2- In the introduction, authors are trying first to expose why GID symptoms are have to be managed in patients with anorexia nervosa. They provide several studies without really making their point clear. For example, it is not clear how GID symptoms are associated with poor body image. It is not clear how much patients with GID have poor treatment adherence and hospital admission as compared with patients without GID.

Thank you for these important suggestions. We have added additional detail to the introduction to clarify the impact of GID on poor body image, poor treatment adherence and lower treatment success [pg 2, lines 53-58]. Unfortunately, as data on GID are not routinely collected and reported, differences in treatment outcomes for those with and without GID have not been explored. However, this is a very important question, and we believe this should be investigated as a priority. We have highlighted this point in pg2, lines 64-66 and lines 89-90.

3- I think that one of the goals for the introduction is to provide the reader with information about GID symptoms especially regarding what are their causes and how to manage them. Authors have named 3 causes for GID symptoms (IBS, abdominal pain and constipation) and provided a brief dietetic management for IBS and constipation. However, the idea is still mending. It is important to be more authoritative and clearer about the main causes of GID symptoms and what should be ususally suggested for patients with anorexia nervosa and GID symptoms.

Thank you for highlighting additional information needed in the introduction. Further information about the complex nature of GID and AN and possible causes have been added [pages 1-2, lines 42-47].

Additional information about the management of GID has been added to the introduction, including suggested medications and emerging therapeutics [pg 2, lines 58-64].

4- Authors may have access to data related to Australian dieticians' age, gender, years of experience, location of practice, etc. Accordingly, a comparison between the group of responders with the group of Australian dieticians in relation to some sociodemographic parameters may give the study more power.

Participant demographics are presented in Table 1 and demographics of dietitians in the higher and lower knowledge groups are compared. Differences are observed only for the proportion of dietitians working in private practice, the proportion of dietitians receiving formal training in treating AN, and the proportion of dietitians receiving formal training in treating co-occurring AN and GI. A comparison between the sample of dietitians recruited for this study and the broader workforce of Australian dietitians has been added to the strengths and limitations section [page 10, lines 592-600].

5- The ANGI-KAPS was not rightly described. First, the knowledge part needs to be explicated. The reader may need to know what are the questions asked in the instrument and to develop a personal judgment on it. Second, the tool's validity needs to be better discussed.   

Thank you for these suggestions to strengthen the description of the ANGI-KAPS tool. Firstly, the tool was included as an appendix so that the reader can review the tool in full (Supplementary File 1). Secondly, we have provided more detail about the validation of the tool [page 3, lines 360-373].

Reviewer 2 Report

This manuscript reports on a survey of dieticians about their knowledge, attitudes and practices regarding GID in patients with eating disorders. This is an important and novel question.

Inclusion criteria should be presented more clearly. “Participants were eligible for registration as an Accredited Practicing Dietitian (APD), and had current or recent experience treating patients with a diagnosis of AN.” This sentence is unclear to me. Did study participants need to be registered? Or eligible to be registered?

Survey Instrument development is an important part of this report. As it is not described in more detail elsewhere, please add more detail about the process of evaluating validity and the results of this process. Please describe the level of expertise of the panel in greater detail (number of people, years of experience), the framework used for discussion, reaching consensus. Were any suggestions made and were they adopted?

Also, the limitation section should include discussion of the limitations of the questionnaire which was developed for the purpose of this study.

Author Response

1 - Inclusion criteria should be presented more clearly. “Participants were eligible for registration as an Accredited Practicing Dietitian (APD), and had current or recent experience treating patients with a diagnosis of AN.” This sentence is unclear to me. Did study participants need to be registered? Or eligible to be registered?

It is possible to practice as a dietitian without being registered as an Accredited Practising Dietitian (the Australian credentialing system). Therefore, the inclusion criteria for this study were that participants needed to be eligible for registration as an APD, i.e. have completed the required training to be an APD.

2 - Survey Instrument development is an important part of this report. As it is not described in more detail elsewhere, please add more detail about the process of evaluating validity and the results of this process. Please describe the level of expertise of the panel in greater detail (number of people, years of experience), the framework used for discussion, reaching consensus. Were any suggestions made and were they adopted? 

Thank you for this suggestion, we understand that additional information about the development of the ANGI-KAPS tool will strengthen the manuscript. We have added information to the methods section [page 3, lines 360-374].

3 - Also, the limitation section should include discussion of the limitations of the questionnaire which was developed for the purpose of this study.

We have added limitations of the survey to page 10, lines 592– 597.

Round 2

Reviewer 1 Report

The manuscript has improved. I have no further comments